# Epigenetics in *LMNA*-Related Cardiomyopathy

**DOI:** 10.3390/cells12050783

**Published:** 2023-03-01

**Authors:** Yinuo Wang, Gergana Dobreva

**Affiliations:** 1Department of Cardiovascular Genomics and Epigenomics, European Center for Angioscience (ECAS), Medical Faculty Mannheim, Heidelberg University, 68167 Mannheim, Germany; 2German Centre for Cardiovascular Research (DZHK), 68167 Mannheim, Germany

**Keywords:** nuclear lamina, lamin A/C, *LMNA*, cardiomyopathy, epigenetics, chromatin architecture, stem cells

## Abstract

Mutations in the gene for lamin A/C *(LMNA)* cause a diverse range of diseases known as laminopathies. *LMNA*-related cardiomyopathy is a common inherited heart disease and is highly penetrant with a poor prognosis. In the past years, numerous investigations using mouse models, stem cell technologies, and patient samples have characterized the phenotypic diversity caused by specific *LMNA* variants and contributed to understanding the molecular mechanisms underlying the pathogenesis of heart disease. As a component of the nuclear envelope, *LMNA* regulates nuclear mechanostability and function, chromatin organization, and gene transcription. This review will focus on the different cardiomyopathies caused by *LMNA* mutations, address the role of *LMNA* in chromatin organization and gene regulation, and discuss how these processes go awry in heart disease.

## 1. Introduction

Mutations in genes encoding proteins of the nuclear lamina result in wide-ranging clinical phenotypes collectively referred to as laminopathies [1]. Many of these diseases are caused by mutations in the gene for lamin A/C (*LMNA*) and affect primarily the muscles, the peripheral nerves, and the adipose tissue or cause systemic diseases such as premature aging syndromes [2]. The *LMNA* gene encodes A-type lamins, generated by alternative splicing, of which lamins A and C are the main splicing products [3,4]. In addition to the A-type lamins, the nuclear lamina is composed of B-type lamins, i.e., lamins B1 and B2, encoded by *LMNB1* and *LMNB2* genes, respectively [5,6,7,8]. *LMNB2* also encodes the germ-line-specific lamin B3, produced by alternative splicing [9].

A- and B-type lamins have a common structural organization: a short “head” domain at the N-terminus followed by a central helical rod domain and a C-terminal “tail” domain. The central rod domain is composed of four coiled-coil regions that allow lamins to form parallel coiled-coil dimers and higher-order meshworks [10,11,12]. The “tail” consists of a globular region, which adopts an immunoglobulin (Ig)-like β-fold involved in protein–protein interactions. Pre-lamin A- and B-type lamins also have a CaaX motif at the C-terminus which guides protein farnesylation and carboxyl methylation, important for targeting to the nuclear envelope [10,11,12] (Figure 1).

Both A- and B-type lamins form separate but interconnected filamentous meshworks located between the inner nuclear membrane and the peripheral heterochromatin, which on the one hand provide structural support to the nucleus and on the other hand anchor chromatin at the nuclear periphery, thereby shaping the higher-order chromatin structure [13,14,15]. In contrast to lamins B1 and B2, which are localized at the periphery and associate mainly with transcriptionally inactive chromatin [16,17], lamins A and C are also found in the nuclear interior and associate with both heterochromatin and euchromatin [18]. In addition, lamins interact with the LINC complex, which couples the nucleoskeleton with the cytoskeleton [19,20], and thereby can directly translate mechanical cues and changes in extracellular matrix mechanics into alterations in chromatin structure and transcriptional activity [21]. 

In the last years, a large number of studies identified distinct molecular pathways dysregulated in patients with pathogenic *LMNA* mutations, mouse models, or stem cells carrying *LMNA* mutations. Here, we summarize the current knowledge on the role of lamin A/C in diseases of the heart muscle and specifically focus on how changes in lamin-A/C-dependent chromatin architecture could be involved in the pathogenesis of cardiomyopathies. 

## 2. *LMNA*-Related Dilated Cardiomyopathy

Dilated cardiomyopathy (DCM) is characterized by enlargement and dilatation of one or both ventricles of the heart, which occurs together with impaired contractility and heart function [22]. The *LMNA* gene is the second most commonly mutated gene in familial dilated cardiomyopathy (DCM), accounting for 5% to 8% of cases [23]. Patients carrying pathogenic *LMNA* mutations have a poor prognosis due to the high rate of sudden cardiac death resulting from malignant arrhythmias. Atrial fibrillation (AF), atrioventricular (AV) conduction block, ventricular tachycardia, and sudden cardiac death often precede the development of systolic dysfunction [24,25,26]. Although *LMNA*-related DCM is an adult-onset disease, it cannot be excluded that structural changes and arrhythmias may be present in early asymptomatic individuals [27]. 

To date, around 500 mutations and 300 protein variants have been reported for *LMNA*; detailed information on the different mutations is available through the UMD-*LMNA* mutation database (www.umd.be/*LMNA*, accessed on 3 January 2023) (Table 1). Most of the mutations associated with cardiomyopathies are located in the head and rod domains and are mostly truncation or missense mutations [28]. Heterozygous truncation mutations often result in lamin A/C haploinsufficiency due to a premature termination codon generated by insertions or deletions resulting in a frameshift, aberrant splice site, or nonsense mutations. A homozygous *LMNA* nonsense mutation (Y259X) has also been reported, resulting in a lethal phenotype [29]. *LMNA* missense mutations, on the other hand, are thought to mostly act through a dominant negative mechanism [28]. Patients carrying heterozygous mutations in *LMNA* in combination with mutations within other genes such as *TTN, DES, SUN1/2*, etc., display a particularly severe clinical cardiac phenotype [30,31,32,33,34].

Since a number of *LMNA* mutations result in a loss of function, lamin A/C haploinsufficient (*Lmna*+/−) and *Lmna* knockout mice (*Lmna-*/-) have been extensively used to study the molecular mechanisms underlying *LMNA* loss-of-function (LOF) cardiomyopathy (Table 2). *Lmna*−/− mice develop DCM two weeks after birth and die within one month [62,63]. *Lmna*+/− mice are viable and fertile but already at ten weeks of age show AV conduction defects and atrial and ventricular arrhythmias, characteristic for patients with *LMNA* LOF mutations [64]. Cellular characterization revealed that *Lmna* haploinsufficiency results in AV node cardiomyocyte death and altered electrophysiological properties [64]. Furthermore, *Lmna*−/− and *Lmna*+/− cardiomyocytes (CMs) show premature binucleation, cell cycle withdrawal, and abnormal contractility [63,65]. 

Another mutation often used for modeling the *LMNA* LOF mutation is the p.R225X mutation, a nonsense mutation causing premature truncation of both lamin A and lamin C splice isoforms. Patients carrying this pathogenic mutation show early onset of AF, secondary AV block, and DCM [87]. Like *Lmna*−/− mice, homozygote *Lmna* R225X mice also exhibit retarded postnatal growth, conduction disorders, and DCM [36]. Other LOF mutations, e.g., K117fs and 28insA, also lead to a DCM phenotype. *LMNA* p. K117fs mutation is a frameshift mutation that leads to a premature translation-termination codon [38], whereas 28insA is an adenosine insertion mutation in exon 1 resulting similarly in a premature stop codon [40]. Messenger RNAs (mRNAs) that contain a premature stop codon often undergo degradation through the nonsense-mediated mRNA decay (NMD) surveillance mechanism and thus can cause haploinsufficiency. Consistent with this, a significant decrease in lamin A/C protein levels is observed in K117fs iPSC-CMs as a result of NMD-mediated degradation of *LMNA* mRNA [38]. In addition to truncation mutations, which can result in *LMNA* haploinsufficiency, mutations such as N195K, T10I, R541S, and R337H also show reduced lamin A/C protein levels [35,41]. Patients carrying these pathogenic mutations also develop DCM [26,51,88]. It is still unclear why these mutations lead to decreased lamin A/C levels. Possible reasons could be that protein translation or the stability of lamin A/C are affected in mutant CMs. For example, although *Lmna* mRNA does not change, both lamin A and lamin C levels are decreased in CMs and MEFs derived from *Lmna* N195K/N195K mutant mice [35]. Interestingly, patients carrying different *LMNA* missense mutations resulting in DCM also exhibit lower protein levels [89]. To what extent the decrease in lamin A/C levels or changes in protein function result in disease pathogenesis is still largely unknown and needs further investigation. 

Although it may seem that DCM is predominantly caused by *LMNA* haploinsufficiency, missense mutations in *LMNA*, which do not lead to changes in lamin A/C protein levels, also result in DCM. For example, *LMNA* K219T missense mutation causing severe DCM and heart failure with conduction system disease [52] does not lead to obvious changes in lamin A/C levels in K219T iPSC-CMs [53]. *LMNA* H222P missense mutation has been shown to cause Emery–Dreifuss muscular dystrophy (EDMD) and DCM in patients. Homozygous mice with the H222P mutation display muscular dystrophy, left ventricular dilatation, and conduction defects and die by 9 months of age [76]. Similarly to the K219T mutation, Western blot analysis of cardiac and skeletal muscle samples shows no obvious difference in lamin A/C protein levels between wild-type and *Lmna* H222P/H222P mice [74]. Interestingly, recent studies suggested a developmental origin of *LMNA*-related cardiac laminopathy. *Lmna* H222P/H222P embryonic hearts showed noncompaction, dilatation, and decreased heart function already at E13.5[75], while *Lmna*+/− and *Lmna*−/− embryonic hearts showed noncompaction cardiomyopathy with no decrease in ejection fraction [63]. Differentiation of mouse embryonic stem cells (ESCs) harboring the *Lmna* p.H222P mutation revealed decreased expression of cardiac mesoderm marker genes, such as *Eomes* and *Mesp1* as well as cardiac progenitor (CP) markers and impaired CM differentiation. This is in stark contrast to *Lmna*+/− and *Lmna*−/− mESCs, which showed premature CM differentiation [63,75], suggesting different mechanisms behind the heart phenotype caused by lamin A/C haploinsufficiency or changes in protein functionality. 

Among laminopathy-associated missense mutations, the addition of proline is the most common. Proline addition can significantly alter protein structure. For example, *LMNA* S143P missense mutation causes DCM and disturbs the coiled-coil domain, thus affecting lamin A/C assembly into the nuclear lamina. This results in nuclear fragility and reduced cellular stress tolerance [49]. The addition of proline might also affect protein phosphorylation through proline-directed kinases, such as the mitogen-activated protein (MAP) kinases, cyclin-dependent protein kinase 5 (CDK5), glycogen synthase 3, etc. Mutations resulting in the addition of proline often result in striated muscle disease, suggesting a common underlying mechanism [90].

## 3. Arrhythmogenic Right Ventricular Cardiomyopathy

Arrhythmogenic right ventricular cardiomyopathy (ARVC) is an inherited heart muscle disorder that predominantly affects the right ventricle [91]. A progressive loss of myocytes and fibro-fatty replacement associated with arrhythmias in the right ventricular myocardium is a hallmark of the disease [92]. Mutations in desmosomal genes, such as Plakophilin 2 (*PKP2*), Desmoplakin (*DSP*), Desmoglein 2 (*DSG2*), Desmocollin 2 (*DSC2*), and junction plakoglobin (*JUP*), are the main cause of ARVC [93,94,95,96,97,98]. In addition, mutations in the calcium-handling protein Ryanodine Receptor 2 (*RYR2*) [99], Phospholamban (*PLN*) [100], the adherens junction protein Cadherin 2 (*CDH2*) [101], Integrin-Linked Kinase (*ILK*) [102], the signaling molecule Transforming Growth Factor-β3 (*TGFB3*) [103], the cytoskeletal structure protein Titin (*TTN*) [104], Desmin (*DES*) [105], transmembrane protein 43 (*TMEM43*), and lamin A/C (*LMNA*) have also been reported in ARVC [24,106,107,108].

In 2011, Quarta et al. first reported ARVC caused by mutations in *LMNA*. Four *LMNA* variants were identified: R190W, R644C, R72C, and G382V [24]. The R190W and R644C variants also cause DCM and left ventricular noncompaction (LVNC). In addition, R644C can also lead to lipodystrophy and atypical progeria, thus showing an extreme phenotypic diversity. ARVC patients with these four mutations all exhibit RV dilatation and systolic dysfunction. Histological examination of the right ventricular myocardium from R190W and G382V patients showed a loss of more than 50% of myocytes and extensive interstitial fibrosis and fatty replacement [24]. Interestingly, immunohistochemical staining showed significantly reduced plakoglobin expression at the intercalated discs in the myocardium, which could contribute to the development of ARVC [24]. M1K, W514X, and M384I mutations in *LMNA* have also been identified in ARVC. Patients with M1K and W514X mutations show RV dilatation, non-sustained ventricular tachycardia, and complete atrioventricular block [108]. A patient with the M384I variant not only developed ARVC but also peripheral neuropathy and peroneal muscular atrophy [109]. 

So far, it remains unknown how *LMNA* mutations result in ARVC. Since *LMNA* is a ubiquitously expressed protein, its mechanoprotective function in cardiomyocytes, which can limit the progressive loss of myocytes, its role in the regulation of genes involved in cardiac contractility, and its important role in regulating cell fate choices, which may result in an excess of fibroblasts and adipocytes, might be involved. Tracing back the origins of fat tissue in a mouse model of ARVC, Lombardi et al. suggested that second heart field (SHF)-derived progenitor cells switch to an adipogenic fate through nuclear plakoglobin (JUP)-mediated Wnt signaling inhibition [110]. A subset of resident cardiac fibro-adipocyte progenitor cells characterized by PDGFRA^pos^Lin^neg^THY1^neg^DDR2^neg^ expression signatures have been shown to be a major source of adipocytes in ARVC caused by Desmoplakin (*DSP*) haploinsufficiency [111]. Furthermore, the endocardium, epicardium, and cardiac mesenchymal stromal cells also serve as a source of adipocytes in the heart [112,113,114]. Because the endocardium and epicardium give rise to diverse cardiac cell lineages, including mesenchyme and adipocytes [115], via endothelial-to-mesenchymal transition (EndMT) and epithelial-to-mesenchymal transition (EMT), lamin A/C function in regulating EMT [75] might also be a key mechanism driving ARVC pathogenesis. 

## 4. Left Ventricle Noncompaction Cardiomyopathy

Left ventricular noncompaction (LVNC) cardiomyopathy is a rare congenital heart disease resulting from abnormal development of the endocardium and myocardium. Patients with LVNC exhibit a thin compact myocardium and excessive trabeculation and can eventually develop progressive cardiac dysfunction followed by heart failure. LVNC can manifest together with other cardiomyopathies and congenital heart disease [116]. Studies have identified various genes associated with LVNC, such as *TTN*, *MYH7*, *TNNT2*, *LDB3*, *MYBPC3*, *ACTC1*, *DSP*, *CASQ2*, *RBM20*, and the intermediate filaments *DES* [117] and *LMNA* [118], with the two most affected genes being *TTN* and *LMNA* [119]. The first reported *LMNA* mutant variant causing LVNC is R190W, which is also associated with familial DCM and ARVC [56]. Another pathogenic *LMNA* variant causing LVNC is *LMNA* R644C. R644C mutation carriers show an extreme phenotypic diversity, ranging from DCM and LVNC to lipodystrophy and atypical progeria [59]. Parents and colleagues reported four family members with the *LMNA* R644C mutation, three of whom developed left ventricular noncompaction cardiomyopathy with normal LV dimensions and function and without evidence of dysrhythmias [60]. Other mutations such as *LMNA* V74fs, R572C, and V445E have also been associated with LVNC. Patients with the V445E missense mutation are characterized by an arrhythmogenic form of LVNC, suggested to be due to dysfunctional SCN5A [58,119]. 

How *LMNA* mutations result in LVNC and the mechanisms underlying the high phenotypic diversity are largely unknown. Two recent studies demonstrated that *Lmna* H222P/H222P as well as *Lmna*−/− and *Lmna*+/− embryonic hearts exhibit noncompaction, suggesting these mouse models as important tools to study the developmental origin and the mechanisms behind *LMNA*-mediated noncompaction cardiomyopathy [63,75]. Interestingly, our own study revealed that *Lmna* LOF results in abnormal cell fate choices during cardiogenesis, i.e., promotes CM and represses endothelial cell fate. Since the crosstalk between CMs and endothelial cells is instrumental for proper cardiac development and myocardial compaction [120], abnormal cardiovascular cell fate choices and dysfunctional endothelium might also contribute to LVNC. Thus, understanding the link between alternative cell fate choices, changes in cell behavior, and tissue-specific phenotypes caused by pathogenic *LMNA* mutations would be an important question to address in further studies.

## 5. Restrictive Cardiomyopathy

Restrictive cardiomyopathy (RCM) is a rare cardiac disease characterized by increased myocardial stiffness resulting in impaired ventricular filling. Patients with RCM show enlarged atria and diastolic dysfunction, while systolic function and ventricular wall thicknesses are often normal until the later stages of the disease [121,122,123]. Although most causes of RCM are acquired, several gene mutations have also been identified in patients with RCM [121,122,123,124]. The most common mutated genes found in RCM are sarcomere-related genes such as *TTN* [125], *TNNI3* [126], *MYH7* [127], *ACTC1* [128], etc. Mutations in non-sarcomere genes such as *DES* [129], *TMEM87B* [130], *FLNC* [131], etc., have also been reported. Recently, Paller et al. reported a truncation mutation of *LMNA* (c.835 delG:p.Glu279ArgfsX201) in an RCM patient who had a significant biatrial enlargement, atrial fibrillation, and skeletal muscle weakness. Both right and left ventricular size and function were normal, and histological analysis revealed cardiac hypertrophy and focal interstitial fibrosis in the endomyocardial tissue [61]. How *Lmna* mutations cause RCM is not known; a plausible mechanism could be the activation of profibrotic signaling, as discussed below.

## 6. Molecular Mechanisms Resulting in *LMNA*-Related Cardiomyopathy Pathogenesis

Since *LMNA*-related cardiomyopathies caused by distinct point mutations show phenotypic diversity, the precise molecular mechanisms resulting in disease pathogenesis are also distinct and complex. Taking into account the variety of different functions of the nuclear lamina, three central mechanisms have been suggested to drive disease pathogenesis. 

The “mechanical hypothesis” proposes that disruption of the nuclear lamina causes increased nuclear fragility and increased susceptibility to mechanical stress [132]. This hypothesis is supported by observations that CMs from patients or mouse models with lamin A/C mutations exhibit nuclear rupture, DNA damage, and cell cycle arrest [63,65,88,133]. Interestingly, *Lmna*−/− non-CMs subjected to stretch show significantly increased DNA damage, further supporting the notion that the elevated cell death could be due to the inability of *Lmna*−/− CMs to respond adequately to mechanical stress [63]. Importantly, a recent study revealed that disrupting the LINC complex and thereby decoupling the nucleus/nucleoskeleton from the mechanical forces transduced by the cytoskeleton increases more than fivefold the lifespan of *LMNA*-deficient mice [134], pointing to therapeutic opportunities for patients carrying mutations resulting in nuclear fragility. 

Myriad studies have demonstrated a role of lamins in regulating MAPK, TGF-β, Wnt–β-catenin, and Notch signaling cascades [135,136] and suggested that altered signaling is a key driver of *LMNA-*related dilated cardiomyopathy. For instance, *LMNA*-related cardiomyopathy shows a significant increase in myocardial fibrosis which contributes to left ventricular dysfunction and heart failure [24,35,137,138]. Profibrotic signaling, such as TGF-β, MAPK, and ERK signaling, is activated in *Lmna* H222P/H222P mice, and the partial inhibition of ERK and JNK signaling before the onset of cardiomyopathy in *Lmna* H222P/H222P mice significantly reduces cardiac fibrosis and prevents the development of left ventricle dilatation and decreased cardiac ejection fraction [138,139,140,141]. Indeed, therapies targeting intracellular signaling alterations are being developed in a preclinical setting [142]. 

Since nuclear lamins anchor chromatin at the nuclear periphery, the “chromatin hypothesis” suggests that chromatin alterations as a result of *LMNA* haploinsufficiency or mutation result in abnormal gene expression programs responsible for the disease phenotype [132]. In the last years, a number of studies using iPSC-CMs or mESC-CMs uncovered changes in chromatin architecture coupled to transcriptional changes in different ion channels such as *SCN5A*, *CACNA1A/C/D*, *HCN4*, *SCN3b*, and *SCN4b*, as well as *Pdgfb* pathway activation, which might explain the arrhythmogenic conduction defects in *LMNA* patients [38,53,63,143]. 

## 7. Epigenetics in *LMNA*-Related CARDIOMYOPATHIES

Lamina-associated domain reorganization and changes in chromatin architecture in *LMNA*-related cardiomyopathy.

As already mentioned, the nuclear lamina shapes higher-order chromatin structure by anchoring large heterochromatic regions (~ 0.1–10 Mb stretches) at the nuclear periphery, termed lamina-associated domains (LADs). LADs are enriched in the repressive histone marks H3K9me2/3 and H3K27me3, and genes associated with LADs are mostly inactive [15]. Although most LADs are conserved between cell types (constitutive LADs (cLADs)), some chromatin nuclear lamina interactions are detected in specific cell types (facultative LADs (fLADs)) (Figure 2) [144,145]. Indeed, genome–nuclear lamina dynamics have been proposed to play a key role in cell fate decisions by “locking” or “unlocking” genes conferring cell identity at the nuclear periphery [145,146]. For example, during mESC differentiation into astrocytes (ACs), specific AC genes detach from ESC LADs resulting in gene activation. A substantial number of genes are not immediately activated upon detachment from the nuclear lamina but rather become primed for activation at a later stage [145]. Similar mechanisms also occur during CM differentiation. HDAC3 directly represses cardiac differentiation by tethering CM genes to the nuclear lamina. The loss of HDAC3 in cardiac progenitor cells releases these genomic regions from the nuclear periphery, leading to early cardiac gene expression and differentiation [147]. Our own study further showed that lamin A/C and not B-type lamins is responsible for the early activation of a transcriptional program promoting CM versus endothelial cell fate and differentiation [63]. Interestingly, lineage shifts upon *LMNA* loss or mutation have been reported in other tissues, suggesting that aberrant activation of genes driving an unscheduled differentiation could be a common feature of laminopathic cells [148,149,150,151]. Similar to ACs, we found two modes of regulation: (i) Lamin A/C keeps cell differentiation and cardiac morphogenesis genes silent, such as *Gata4*/6, *Bmps*, *Wnts*, *Myl*4, etc. Upon lamin A/C LOF, these genes are ectopically expressed in mESCs. (ii) Lamin A/C restricts transcriptional permissiveness at cardiac structural and contraction genes, such as *Ryr2*, *Mybpc3*, *Adrb2*, etc. Upon lamin A/C LOF, chromatin becomes more accessible, but this is not sufficient to elicit gene transcription in ESCs. However, during cardiac differentiation, these primed loci are readily accessible to cardiac transcription factors (TFs), resulting in aberrant cardiovascular cell fate choices, premature CM maturation, cell cycle withdrawal, and abnormal contractility. In contrast, *Lmna* H222P/H222P mESCs, or mESCs harboring the G609G mutation causing accelerated aging, did not show similar changes in chromatin accessibility nor in expression patterns, supporting the view that the molecular mechanisms underlying the distinct phenotypes upon lamin A/C LOF and missense mutations are different [63].

Many recent studies have focused on the role of lamin A/C in chromatin organization in human induced pluripotent stem cell (hiPSC)-derived CMs (hiPSC-CMs) to pinpoint the molecular mechanisms associated with *LMNA* cardiomyopathy. For instance, in hiPSC-CMs harboring the frameshift mutation K117fs that leads to lamin A/C haploinsufficiency, chromatin accessibility is increased at lamin A/C LADs, leading to transcriptional activation. Among others, the PDGF pathway was highly activated in K117fs iPSC-CMs and its inhibition rescued the arrhythmic phenotype, suggesting that PDGF inhibitors could be beneficial in preventing fatal arrhythmias often manifested in patients with *LMNA*-related cardiomyopathy [38]. Notably, the authors found that many genes located in non-LAD regions are also highly upregulated in K117fs iPSC-CMs compared to controls, suggesting that mutations in lamin A/C might also result in maladaptive epigenetic remodeling at non-LAD regions. This might be mediated through changes in B-type lamin function, upregulation of pioneer transcription factors, loss of binding of repressive complexes, or other mechanisms. Indeed, although B-type lamins form distinct meshworks, the loss of A-type lamins results in alterations in B-type meshworks, suggesting that their activity might be interconnected [152]. Thus, mutation-mediated changes in lamin A/C activity might also affect lamin B1/B2 function. Interestingly, lamin B2 plays an essential function in regulating CM karyokinesis, and *Lmnb2* ablation resulted in polyploid CM nuclei in neonatal mice [153]. *Lmna* ablation also results in increased numbers of binucleated CMs in neonatal mice [63], suggesting that lamin A/C loss might affect lamin B2 function. The activation of pioneer transcription factors, which can engage developmentally silenced genes embedded in “closed” chromatin [154,155,156,157] and induce chromatin opening, might also play a role in *LMNA*-related cardiomyopathies. Indeed, the pioneer cardiac TF GATA4 is activated by lamin A/C loss, and *Gata4* silencing or haploinsufficiency rescues the abnormal cardiovascular cell function induced by lamin A/C deficiency [63]. Another pioneer TF, FoxO1 [158], also shows increased binding to chromatin upon *Lmna* LOF. FoxO TFs play key functions in stress response, cell proliferation, and apoptosis, and the longevity and suppression of FoxO activity in CMs partially rescues the cardiac phenotype and prolongs survival [159]. Additionally, the nuclear lamina may serve as a binding platform for chromatin remodelers, such as the Polycomb Group Proteins, which can initiate large-scale epigenetic alterations. This will be discussed in the following section (Figure 3).

Another study using an iPSC model harboring the T10I mutation in *LMNA* suggested a role of the nuclear lamina in safeguarding cellular identity [41]. In T10I iPSC-CMs, the peripheral heterochromatin enriched for non-myocyte lineage genes was disrupted, resulting in the activation of alternative cell fate genes. Upregulation of non-cardiac genes was also observed in iPSC-CMs carrying the R225X mutation in lamin A/C (Figure 2B). Importantly, *CACNA1A*, encoding a neuronal P/Q-type calcium channel, was upregulated, and pharmacological inhibition partially rescued the altered electrophysiological properties of R225X iPSC-CMs [143]. In this context, it is important to note that in contrast to mouse/human blastocysts and naïve mouse mESCs, hiPSCs cultured in standard conditions represent a primed state and do not express detectable levels of lamin A/C protein [63]. Since lamin A/C plays an important role in chromatin organization in naïve pluripotent stem cells, which is essential for normal cardiogenesis, some important aspects of lamin A/C function cannot be modeled using hiPSCs and requires studies using naïve hiPSCs carrying *LMNA* mutations. 

In addition, chromatin and expression analysis of CMs from patients with different *LMNA*-related DCM mutations revealed extensive rearrangement of *LMNA*–chromatin interactions in DCM patients [89]. The reorganization of lamin A/C LADs is associated with altered CpG methylation and dysregulated expression of a large number of genes involved in cell metabolism, the cell cycle, and cell death. Most of the *LMNA*-related DCM patients’ samples used in this study showed a decrease in lamin A/C protein levels, suggesting that *LMNA* LOF might account for the observed DNA, chromatin, and expression changes [89].

It is still poorly understood how cell-type-specific tethering at the nuclear lamina is achieved and how mutations in lamins affect the tethering of key cell fate determinants in stem cells and in cells already committed to a certain lineage. Lamins interact with chromatin either directly [160] or indirectly through chromatin-binding proteins. Consistent with its association with both hetero- and euchromatin, lamin A/C interacts with proteins associated with both hetero- and euchromatin, e.g., LAP2α, Emerin, and BANF1 [161,162], while B-type lamins interact with the lamin B receptor (LBR), which mediates the attachment to the inner nuclear membrane, and Heterochromatin Protein 1 (HP1α) associated with heterochromatin [163]. However, all these proteins are broadly expressed and cannot account for the cell-type-specific tethering of LADs. Thus, identifying cell-type-specific interacting partners for nuclear lamins and the effect of lamin mutations on these interactions will be key in understanding the molecular mechanisms underlying the wide-ranging clinical phenotypes and may pinpoint druggable protein–protein interfaces for therapeutic applications. 

Moreover, how mutations in lamin A/C affect the separation into relatively active and inactive chromatin regions is still debatable [164]. The genome is organized into higher-order structural domains referred to as topologically associated domains (TADs). TADs tend to interact based on their epigenetic status and transcriptional activity, thus dividing chromosomes into two types of large-scale compartments generally called A compartments (active) and B compartments (inactive) (Figure 3A) [165]. An analysis of A/B compartment changes revealed only ∼1.2% compartment switches in R225X iPSC-CMs with only a minimal correlation with highly dysregulated genes [143]. In contrast, during cardiac differentiation, ∼20% of the genome undergoes A/B compartment reorganization, while many others appear constitutively associated with the nuclear lamina. Interestingly, in *Lmna*−/− mESC, around 8% of the chromatin compartments switched from A to B and vice versa as a result of lamin A/C loss. These compartment switches highly overlap with lamin A LADs. Genes within the B/A compartment switches (inactive to active) were linked to calcium ion transmembrane transport, muscle cell differentiation, and relaxation of cardiac muscle, including genes such as *Myl4*, *Atp2a3*, *Ryr2*, and *Camk2d*, which were either activated or primed upon lamin A/C loss. Lamin A/C is expressed in naïve pluripotent stem cells, absent after the loss of pluripotency and during early differentiation, and re-expressed in CMs. This dynamic expression pattern may provide a window of opportunity for LAD and chromatin compartment reorganization, and the activation of transcriptional programs driving important developmental decisions and cell identity.

The role of Polycomb Group Proteins in *LMNA*-related cardiomyopathy.

As we discussed before, LADs are enriched for H3K27me3. The downregulation of lamin A/C remodels the repressive H3K27me3 and the permissive H3K4me3 histone marks, thereby enhancing transcriptional permissiveness [166]. Indeed, lamin A/C interacts with the Polycomb repressive complex 2 (PRC2) complex, which catalyzes H3K27me3 [167], and lamin A/C loss in myoblasts results in PcG protein foci disassembly, ectopic expression of Polycomb targets, and premature myogenic differentiation [167]. Polycomb Group (PcG) proteins are key epigenetic repressors during development and differentiation. The Polycomb repressive complex 2 (PRC2)-mediated deposition of H3K27me3 recruits the canonical Polycomb repressive complex 1 (PRC1) that monoubiquitinates lysine 119 of histone H2A (H2AK119ub1) and induces chromatin compaction. The core PRC2 is formed by EED, SUZ12, and the catalytic components EZH2 or EZH1 (Figure 3B) [168,169]. Both PRC1 and PRC2 play an important role in cardiac development and differentiation. EZH2 is essential for CM proliferation, survival, and postnatal cardiac homeostasis. The inactivation of *Ezh2* specifically in cardiac progenitors results in ectopic transcriptional programs and lethal heart defects [170,171]. PRC2 function also ensures proper cardiac growth, and *Eed* ablation by TnT-Cre leads to myocardial hypoplasia and embryonic lethality [170,171]. In a mouse model of EDMD, lamin A/C loss results in PcG repositioning and de-repression of non-muscle genes in muscle satellite stem cells together with the activation of *p16INK4a* that induces cell cycle arrest. This aberrant transcriptional program causes impairment in self-renewal, loss of cell identity, and premature exhaustion of the quiescent satellite cell pool [172]. In a recent study using iPSC-CMs carrying the cardiac-laminopathy-associated K219T mutation, it was shown that the binding of lamin A/C together with PRC2 at the *SCN5A* promoter represses its expression, resulting in decreased conduction velocity [53]. Together, aberrant PRC activity upon *LMNA* mutation might play an important role in *LMNA*-related cardiomyopathies (Figure 3B).

## 8. Advances in Therapeutic Strategies for *LMNA-*Related Cardiomyopathy 

The clinical management of *LMNA*-related DCM includes pharmacological treatment with ACE inhibitors and beta blockers and implantable cardiac defibrillators (ICDs) [173,174]. Heart transplantation or ventricular assist devices may also be required for patients in the end stages of heart failure [173,174]. The inhibition of mTOR, MAPK, and LSD1 significantly rescues the *LMNA*-related DCM phenotype in mice [75,138,175], and a novel and selective p38 MAPK inhibitor is now in a phase 3 clinical trial in *LMNA*-related DCM [176]. In addition, CRISPR/Cas9-based genome editing strategies have been used in *LMNA*-caused Hutchinson–Gilford Progeria Syndrome (HGPS) and show promising results [177,178,179]. By using guide RNAs (gRNAs) that target *LMNA* exon 11 to specifically interfere with lamin A/progerin expression, both Santiago-Fernández et al. and Beyret et al. show a reduced progerin expression and improvement in the progeria phenotype in an HGPS mouse model [177,178]. However, off-target effects, e.g., resulting from insertion and deletions during non-homologous end joining (NHEJ), are a major concern. To overcome these limitations, CRISPR/Cas9-mediated base pair editing systems have been used in HGPS mice [179]. Base pair editing systems could modify the genome without the need of double-strand DNA breaks or donor DNA templates [180]. Two classes of DNA base editors have been reported: cytosine base editors (CBEs), which convert C:G to T:A, and adenine base editors (ABEs) which convert A:T to G:C [181,182]. Systemic injection of a single dose of dual AAV9 encoding ABE and sgRNA into an HGPS mouse model significantly extends the median lifespan of the mice, improves aortic health, and fully rescues VSMC counts as well as adventitial fibrosis [179]. Despite the power of the base pair editing technology, a major limitation is the inability to edit the genome beyond four transition mutations. Prime editing represents a novel approach which is not only suitable for all transition and transversion mutations but also for small insertion and deletion mutations [183]. Similar to base pair editing, prime editing does not require double-strand DNA breaks or donor DNA templates [183] and could be used in the correction of genetic cardiomyopathies.

## 9. Conclusions and Perspectives

Accumulating evidence shows that epigenetic alterations play a crucial role in *LMNA*-related cardiomyopathies. Mutations in *LMNA* affect 3D genome architecture and chromatin accessibility, thereby altering gene expression programs. Several prospective target genes, such as *PDGFRB*, *Gata4*, *SCN5A*, and *CACNA1A,* have been identified using experimental models harboring different *LMNA* mutations, which may serve as potential therapeutic targets. As reviewed above, specific *LMNA* variants can cause extreme phenotypic diversity, which makes it challenging to understand the primary changes underlying disease pathogenesis and thus to design specific treatment strategies for patients. Therefore, an important question remains: how do different and specific *LMNA* mutations result in phenotypic diversity? Environmental factors, such as diet, exercise, and stress, as well as age, sex, and other comorbidities, might also contribute to the phenotypic variability in patients with pathogenic *LMNA* mutations. Identifying cell-type-specific interacting partners for nuclear lamins and the effect of lamin mutations on these interactions would also be important in understanding the wide-ranging clinical phenotypes and may pinpoint druggable protein–protein interfaces for therapeutic applications. Given the important role of lamin A/C in heart development and CM differentiation, developmental changes in asymptomatic-at-birth *LMNA* patients might result in late changes in heart structure and function, warranting further investigation.

## Figures and Tables

**Figure 1 cells-12-00783-f001:**
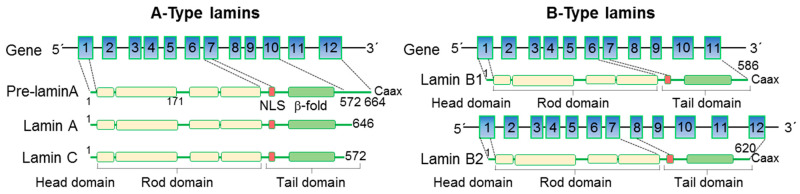
Structure of nuclear lamins. A- and B-type lamins have a conserved domain structure, consisting of a short N-terminal “head” domain, central helical coiled-coil rod domain, and a C-terminal immunoglobulin (Ig)-like β-fold domain. The nuclear localization signal (NLS) is located at the beginning of the tail domain. Pre-lamin A- and B-type lamins also have a CaaX motif at the C-terminus guiding their targeting to the inner nuclear membrane.

**Figure 2 cells-12-00783-f002:**
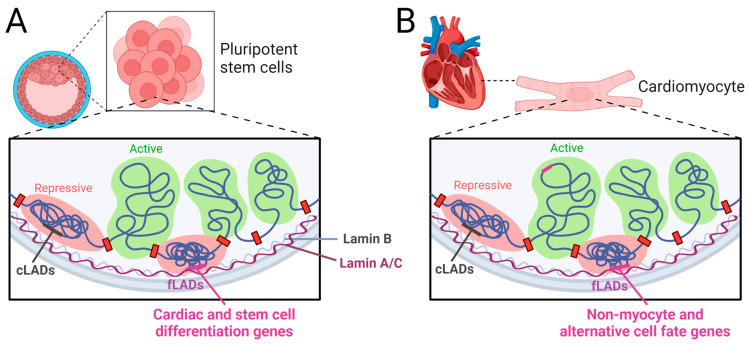
Function of lamin A/C in naïve pluripotent stem cells (**A**) and CMs (**B**). In naïve pluripotent stem cells (**A**), lamin A/C anchors cardiac genes and genes involved in stem cell differentiation to the repressive nuclear periphery. Lamin A/C LOF leads to their detachment from the nuclear lamina accompanied with either immediate activation or epigenetic priming for activation later in cardiogenesis. These chromatin alterations result in premature cardiomyocyte differentiation, cell cycle withdrawal, and abnormal contractility. cLADs—constitutive LADs; fLADs—facultative LADs. In cardiomyocytes (**B**), the nuclear lamina anchors non-myocyte genes and genes involved in alternative cell fates to the nuclear periphery. Mutations in *LMNA* lead to disruption of the peripheral heterochromatin, resulting in activation of alternative cell fate genes. Figures were designed with BioRender.

**Figure 3 cells-12-00783-f003:**
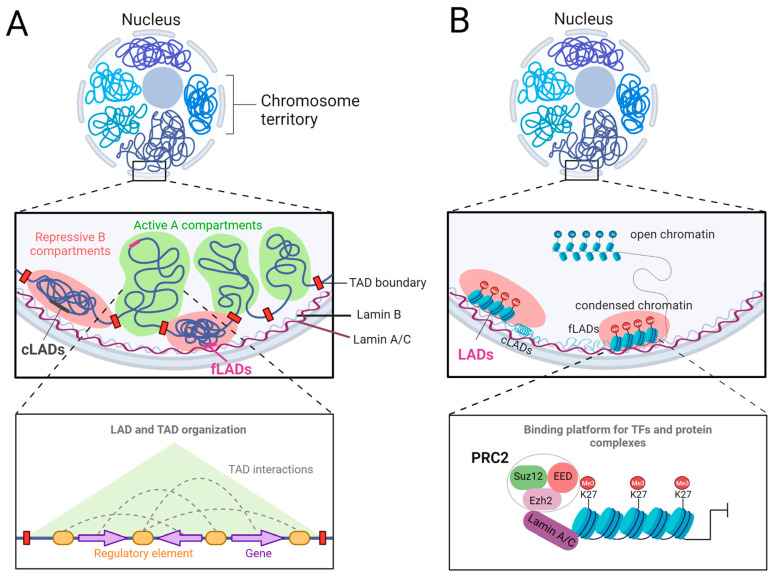
Lamin-A/C-mediated chromatin organization. (**A**) Nuclear lamins play a key role in shaping the 3D chromatin organization by anchoring chromatin at the nuclear periphery. Mutations in *LMNA* might cause changes in LAD and TAD organization, leading to aberrant transcriptional programs. A- and B-type lamins form interconnected networks; thus, *LMNA* mutations might alter lamin B function. (**B**) The nuclear lamina serves as a binding platform of TFs and chromatin modifiers, for example, the PRC2 complex, which might be altered upon *LMNA* mutations.

**Table 1 cells-12-00783-t001:** Cardiomyopathies caused by the most well studied pathogenic *LMNA* mutations in patients. Dilated cardiomyopathy (DCM); left ventricular noncompaction cardiomyopathy (LVNC); arrhythmogenic right ventricular cardiomyopathy (ARVC); restrictive cardiomyopathy (RCM); Emery–Dreifuss muscular dystrophy (EDMD); limb–girdle muscular dystrophy (LGMD).

*LMNA* Mutation	Disease	Clinical Features	References
p. N195K	DCM	Heart dilatation, fibrosis, arrhythmia, sinus bradycardia, atrioventricular conduction block, and atrial arrhythmias	[26,35]
p. R225X	DCM	Atrial fibrillation, complete atrioventricular block, ventricular tachyarrhythmia, and heart failure	[36,37]
p. K117fs	DCM	Atrioventricular block, ventricular tachycardia, atrial fibrillation, arrhythmias at the single-cell level	[38]
p.c.908_909 delCT	DCM	Atrial fibrillation, sick sinus syndrome, dilated cardiomyopathy	[39]
p.28insA	DCM	Dilated cardiomyopathy with conduction defects	[40]
p. T101	DCM, lipodystrophy, atypical progeroid syndrome	Hypertriglyceridemia, diabetes mellitus, insulin resistance, left ventricular myocyte hypertrophy, interstitial fibrosis	[41,42,43,44]
p. R377H	EDMD, LGMD, DCM	Muscular dystrophy, atrial fibrillation or flutter; conduction defects	[45,46,47,48]
p. S143p	DCM	Atrioventricular conduction defects, left ventricular systolic dysfunction and dilatation	[49,50]
p.H222P	EDMD	Muscle weakness, cardiac arrhythmias	[51]
p. K219T	DCM	Heart dilatation, atrial fibrillation, atrioventricular block	[52,53]
p. E161K	DCM	Dilatation, atrial fibrillation, conduction system diseases	[40,54]
p. R541C	DCM, LVNC	Reduced heart contractility, left ventricle dilatation, polymorphic premature ventricular contraction, diffuse ST-T change	[41,55]
p. R190W	DCM, LVNC, ARVC	Left ventricular noncompaction, conduction system defect, abnormal activation of ERK1/2 signaling and sarcomeric disorganization	[56,57,58]
p. R644C	DCM, LVNC, ARVC,	Also leads to lipodystrophy, atypical progeria, phenotypic diversity, and low penetrance associated with the R644C mutation	[59,60]
p. V445E	LVNC	Ventricular tachycardia/fibrillation	[58]
p.c.835 delG:p.Glu279ArgfsX201	RCM	Diastolic dysfunction, biatrial enlargement, atrial fibrillation, skeletal muscle weakness	[61]

**Table 2 cells-12-00783-t002:** Mouse models of laminopathies.

Mouse Model	Gene Targeting Strategy	Disease	Homozygous Phenotype	Heterozygous Phonotype	References
*Lmna*−/− *	Deletion of exons 8-11; a truncated lamin A protein of 54 kDa is still expressed	DCM, EDMD	Retarded postnatal growth, conduction disorders, DCM, EDMD, death by 8 weeks of age	AV conduction defects, both atrial and ventricular arrhythmias; develop DCM by 50 weeks	[64,65]
*Lmna*−/−	Deletion of exon 2	DCM LVNC	Retarded postnatal growth, conduction disorders, DCM, noncompaction, death within 1 month, developmental defects	RV dilatation, RV noncompaction, developmental defects	[62,63]
*Lmna* GT−/−	A gene trap cassette inserted upstream of exon 2 of *Lmna*	N/A	Growth retardation at 2 weeks, impaired postnatal cardiac hypertrophy, skeletal muscle hypotrophy, defects in lipid metabolism	No apparent abnormalities	[66]
*Myh6* cre -*Lmna* f/f	Conditional deletion of *Lmna* in cardiomyocytes	DCM	DCM, cardiac dysfunction, conduction defects, ventricular arrhythmias, fibrosis, apoptosis, and premature death within 4 weeks	Develop cardiac dilatation and dysfunction, cardiac arrhythmias, fibrosis in older mice	[67]
Lamin C only	Deletion of the last 150 nucleotides of exon 11 and the complete intron 11	N/A	No obvious phenotype	No obvious phenotype	[68]
Lamin A only	Deletion of introns 10 and 11, the last 30 bp of exon 11, and the first 24 bp of exon 12	N/A	No apparent abnormalities	N/A	[69]
Pre-lamin A only	Deletion of intron 10	N/A	No apparent abnormalities	N/A	[69]
*Lmna* N195K	Missense mutation in exon 3	DCM	DCM, conduction defects, fibrosis, minor growth retardation, increased heart weight, death at 12–14 weeks	No obvious phenotype	[35]
*Lmna* R225X	Nonsense mutation at exon 4 causing premature truncation of both lamin A and lamin C	DCM	Retarded postnatal growth, conduction disorders, dilated cardiomyopathy, AV node fibrosis, death within postnatal 2 weeks	No apparent abnormalities	[36]
*Lmna* E82K	Transgenic mice expressing *Lmna* E82K under the control of α-MHC promoter	DCM	N/A	DCM, conduction defects, fibrosis, increased heart weight	[70]
*Lmna* ∆K32	Deletion of lysine 32 of lamin A/C in exon 1	DCM	Retarded postnatal growth, striated muscle maturation delay, metabolic defects including reduced adipose tissue and hypoglycemia, death within 3 weeks	Develop a progressive cardiac dysfunction and DCM	[71,72,73]
*Lmna* R541C	Missense mutation in exon 10	DCM	Ventricular dilatation and reduced systolic function	N/A	[55]
*Lmna* H222P	Missense mutation in exon 4	EDMD DCM	Heart dilatation, conduction defects, increased fibrosis, hypertrophy defects, death by 9 months of age, developmental defects	No apparent abnormalities	[74,75,76]
*Lmna* M317K	Transgenic mice expressing *Lmna* M317K missense mutation under the control of α-MHC promoter	EDMD	N/A	Increased eosinophilia and fragmentation of cardiomyofibrils, nuclear pyknosis and edema without fibrosis or significant inflammation, death at 2–7 weeks of age	[77]
*Lmna D300N*	Transgenic mice expressing *Lmna D300N*; Myh6-tTA mice	DCM	N/A	Heart dilatation, increased heart-to-body-weight ratio, fibrosis, death within two months	[78]
*Lmna* L530P	Missense mutation in exon 9	HGPS	Loss of subcutaneous fat, reduction in growth rate, and death by 4 weeks of age	No apparent abnormalities	[79]
*Lmna* G609G	Point mutation in exon 11	HGPS	Shortened life span, reduced body weight, bone and cardiovascular abnormalities, death at an average of 100 days	Develop a similar phenotype to homozygotes but at an older age, average death at 242 days	[80]
*Lmna* HG	Deletion of introns 10–11 and last 150 nucleotides of exon 11	HGPS	Growth retardation osteoporosis, micrognathia, loss of adipose tissue, death by 3–4 weeks of age	Similar phenotype to homozygotes but less severe, death by 21 weeks	[81]
*Lmna* nHG	Deletion of introns 10 and 11 and the last 150 bp of exon 11 together with an exchange of cysteine to serine in the CaaX motif	HGPS	Weight loss, reduced subcutaneous and abdominal fat by 4–8 weeks of age, death at 17 weeks of age	Similar spectrum of disease phenotypes as Lmna HG/+ mice but less severe, death by 36 weeks of age	[82]
csmHG	Deletion of introns 10 and 11 and the last 150 bp of exon 11, three-nucleotide deletion (the isoleucine in progerin’s CaaX motif)	HGPS	No bone phenotype, normal body weight and survival	No bone disease, normal body weight and survival	[83]
G608G BAC	164-kb BAC carrying mutated G608G human *LMNA*	HGPS	N/A	Progressive loss of vascular smooth muscle cells	[84]
tetop_LAG608G	Targeted expression of the lamin A G608G minigene using the keratin 5 transactivator	HGPS	N/A	Growth retardation, skin and teeth abnormalities, fibrosis, loss of hypodermal adipocytes	[85]
Keratin14-progerin	Vector expressing progerin in epidermis under the control of the keratin 14 promoter	HGPS	N/A	Severe abnormalities in skin keratinocyte nuclei, including nuclear envelope lobulation and decreased nuclear circularity	[86]

*: Although initially this mouse model has been used as a *Lmna* knockout mouse model, later studies revealed that a truncated lamin A protein of 54 kDa is still expressed.

## Data Availability

No new data were created.

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
