# Peer review of "Epigenetics in LMNA-Related Cardiomyopathy"

_cells, 2023, doi:10.3390/cells12050783_

Round 1

Reviewer 1 Report

The interesting review by Yinuo Wang and Gergana Dobreva reports the state of the art knowledge on LMNA-related cardiomyopathy in a concise and straightforward way. 
I suggest a few changes.

- line 26: Not only lamin A and lamin C, but also lamin C2 and lamin A delta 10 are LMNA splicing products.  Please, amend by writing that "the main splicing products of LMNA gene are lamin A and lamin C."

- line 28:  "B-type lamins, i.e. lamin B1 and lamin B2, encoded by LMNB1 28 and LMNB2 genes": please, mention also lamin B3.

line 35:  " lamins A and C are also found in the nuclear interior and associate with 35 euchromatin [14]. " Lamin a/C also associates with heterochromatin, please mention it.

- line 36: lamins interact with the LINC complex, are not part of it. Please, correct.

- line 63 nonsense mutations can be found, but the protein is always expressed, please highlight that absence of lamin A/C is lethal at birth in humans and cite refs.

- line 168: please  mention that double heterozygosity can worsen the LMNA cardiac phenotype and cite genes and important refs: Roncarati et al 2013 (LMNA plus TTN mut), Meinke et al. 2014 (LMNA plus SUN1 or SUN2 mut),  Galata et al 2018 and Maggi et al 2021 (LMNA plus DES mut), Montano et al 2022 (mitochondrial DNA mut).

- line 209: please cite also Bernasconi et al 2018 showing that patients with muscular laminopathy featuring cardiomyopathy accumulated TGFbeta 2 in serum.

- lines 242-244: it is very important that the authors have demonstrated aberrant differentiation-related gene expression associated with LMNA mutations. Please, note other papers supporting this shift from one lineage to another in different tissues, mostly adipose tissue (Oldenburg et al., 2017; Pellegrini et al 2019; Ramirez-Martinez, 2021; Czapiewski et al 2022). It is important to mention that aberrant activation of genes starting an unscheduled differentiation could be a common feature of laminopathic cells. 

-  I SUGGEST TO ADD A FEW LINES ON PATHOGENETIC MECHANISMS RELATED TO FIBROSIS (SEE PAPERS BY THE ANTOINE MUCHIR GROUP), WHICH ARE CRUCIAL AND A FIRST SIGN OF LMNA -LINKED DCM.

Author Response

The interesting review by Yinuo Wang and Gergana Dobreva reports the state of the art knowledge on LMNA-related cardiomyopathy in a concise and straightforward way. 
I suggest a few changes.

We would like to express our sincere gratitude to the reviewer for his/hers thoughtful comments, as well as for the time invested in carefully reviewing our manuscript. The reviewer’s comments are in italics.

- line 26: Not only lamin A and lamin C, but also lamin C2 and lamin A delta 10 are LMNA splicing products.  Please, amend by writing that "the main splicing products of LMNA gene are lamin A and lamin C."

Lines 27-29: We have now clarified this point as suggested by the reviewer: ”The LMNA gene encodes A-type lamins, generated by alternative splicing, of which lamin A and lamin C are the main splicing products [3, 4].

- line 28:  "B-type lamins, i.e. lamin B1 and lamin B2, encoded by LMNB1 28 and LMNB2 genes": please, mention also lamin B3.

Lines 31-32: We have now included the following sentence “LMNB2 also encodes the germ line-specific lamin B3, produced by alternative splicing [9].”.

line 35: " lamins A and C are also found in the nuclear interior and associate with 35 euchromatin [14]. " Lamin a/C also associates with heterochromatin, please mention it.

Lines 45-47: We have now revised this sentence to clarify that Lamin A and C associate with heterochromatin as well: ”In contrast to lamins B1 and B2, which are localized at the periphery and associate mainly with transcriptionally inactive chromatin [16, 17], lamins A and C are also found in the nuclear interior and associate with both heterochromatin and euchromatin [18]”.

- line 36: lamins interact with the LINC complex, are not part of it. Please, correct.

Lines 47-51: We thank the reviewer for this comment; we have now revised the sentence accordingly: „In addition, lamins interact with the LINC complex, which couples the nucleoskeleton with the cytoskeleton [19, 20], and thereby can directly translate mechanical cues and changes in extracellular matrix mechanics, into alterations in chromatin structure and transcriptional activity [21]”.

- line 63 nonsense mutations can be found, but the protein is always expressed, please highlight that absence of lamin A/C is lethal at birth in humans and cite refs.

Lines 82-86: We have now clarified that heterozygous nonsense mutations result in haploinsufficiency, whereas homozygous mutation cause lethality as follows: “Heterozygous truncation mutations often result in lamin A/C haploinsufficiency, due to a premature termination codon generated by insertions or deletions resulting in a frameshift, aberrant splice site, or nonsense mutations. A homozygous LMNA nonsense mutation (Y259X) has also been reported, resulting in a lethal phenotype [29].”

- line 168: please  mention that double heterozygosity can worsen the LMNA cardiac phenotype and cite genes and important refs: Roncarati et al 2013 (LMNA plus TTN mut), Meinke et al. 2014 (LMNA plus SUN1 or SUN2 mut),  Galata et al 2018 and Maggi et al 2021 (LMNA plus DES mut), Montano et al 2022 (mitochondrial DNA mut).

Lines 87-89: We have now highlighted the severity of cardiac phenotype in patients carrying double heterozygous mutations: “Patients carrying heterozygous mutations of LMNA in combination with mutations within other genes such as TTN, DES, SUN1/2 etc. display a particularly severe clinical cardiac phenotype [30-34].”.

- line 209: please cite also Bernasconi et al 2018 showing that patients with muscular laminopathy featuring cardiomyopathy accumulated TGFbeta 2 in serum.

We have now included citation of this important study (now line 270).

- lines 242-244: it is very important that the authors have demonstrated aberrant differentiation-related gene expression associated with LMNA mutations. Please, note other papers supporting this shift from one lineage to another in different tissues, mostly adipose tissue (Oldenburg et al., 2017; Pellegrini et al 2019; Ramirez-Martinez, 2021; Czapiewski et al 2022). It is important to mention that aberrant activation of genes starting an unscheduled differentiation could be a common feature of laminopathic cells. 

Lines 309-311: We thank the reviewer for this comment, we have now highlighted that lineage shifts might be a common mechanism in laminopathies, as suggested by the reviewer: “Interestingly, lineage shifts upon LMNA loss or mutation has been reported in other tissues, suggesting that aberrant activation of genes driving an unscheduled differentiation could be a common feature of laminopathic cells [163-166].”

-  I SUGGEST TO ADD A FEW LINES ON PATHOGENETIC MECHANISMS RELATED TO FIBROSIS (SEE PAPERS BY THE ANTOINE MUCHIR GROUP), WHICH ARE CRUCIAL AND A FIRST SIGN OF LMNA -LINKED DCM.

Lines 271-279: We have now added few sentences on the pathogenetic mechanisms related to fibrosis as follows: “For instance, LMNA-related cardiomyopathy shows a significant increase of myocardial fibrosis which contributes to left ventricular dysfunction and heart failure [35, 124, 152, 153]. Profibrotic signaling, such as TGF-β, MAPK, ERK signaling, are activated in Lmna H222P/H222P mice and partial inhibition of ERK and JNK signaling before the onset of cardiomyopathy in Lmna H222P/H222P mice significantly reduces cardiac fibrosis and prevents the development of left ventricle dilatation and decreased cardiac ejection fraction [153-156]. Indeed, therapies targeting intracellular signaling alterations are being developed in preclinical setting [157].”

We have also added a section on restrictive cardiomyopathy (RCM) characterized by increased myocardial stiffness (lines 236-251).

Reviewer 2 Report

In the review article 'Epigenetics in LMNA-related cardiomyopathy' submitted by Wang and Dobreva to Cells, the authors summaryze the knowledge about LMNA mutations leading to cardiomyopathies. The topic of this review is interesting, however, the review article can be improved at different points.

1.) Please prepare a overview figure about lamin A / C and Lamin B1 and B2. Please discuss important protein domains of these proteins.

2.) Please explain inclusion and exclusion criteria for the LMNA-mutation table. This seems a pretty random selection and it is not complete. Human Gene Mutation Database analysis can be added.

3.).Line 93 ... develop DCM. Please add a citation 

4.) Line 97 ... mutant mice. Please add a citation

5.) Please list a relevant review article about the genetic background of DCM at the beginning of paragraph 2. For example you can cite the following book chapter: Gerull, Brenda, Sabine Klaassen, and Andreas Brodehl. "The genetic landscape of cardiomyopathies." Genetic Causes of Cardiac Disease (2019): 45-91.

6.) Western blot and not western blot (Line 109).

7.) Could you please discuss the affect of proline on protein structure of lamin?

8.) Please add at the beginning of paragraph 3 a relevant review article to give the reader an overview about the genetic background of ARVC --> Gerull, Brenda, and Andreas Brodehl. "Insights into genetics and pathophysiology of Arrhythmogenic cardiomyopathy." Current Heart Failure Reports 18 (2021): 378-390.

9.) Please write all human gene names in Capitals and Italics.

10.) Line 125/126 Please add DSC2 encoding desmocollin-2. You should also insert a citation for DSC2. --> Brodehl, Andreas, et al. "A homozygous DSC2 deletion associated with arrhythmogenic cardiomyopathy is caused by uniparental isodisomy." Journal of Molecular and Cellular Cardiology 141 (2020): 17-29.

11.) You should also add the other non-desmosomal genes including the relevant citations in Line 127/128:

DES (Protonotarios, Alexandros, et al. "The novel desmin variant p. Leu115Ile is associated with a unique form of biventricular Arrhythmogenic Cardiomyopathy." Canadian Journal of Cardiology 37.6 (2021): 857-866.),

LEMD2 (Abdelfatah, Nelly, et al. "Characterization of a unique form of arrhythmic cardiomyopathy caused by recessive mutation in LEMD2." JACC: Basic to Translational Science 4.2 (2019): 204-221.),

PLN (Van Der Zwaag, P. A., et al. "Recurrent and founder mutations in the Netherlands—Phospholamban p. Arg14del mutation causes arrhythmogenic cardiomyopathy." Netherlands Heart Journal 21 (2013): 286-293.)

ILK (Brodehl, A., Rezazadeh, S., Williams, T., Munsie, N. M., Liedtke, D., Oh, T., ... & FORGE Canada Consortium. (2019). Mutations in ILK, encoding integrin-linked kinase, are associated with arrhythmogenic cardiomyopathy. Translational Research208, 15-29.)

and CDH2 (Mayosi, Bongani M., et al. "Identification of cadherin 2 (CDH2) mutations in arrhythmogenic right ventricular cardiomyopathy." Circulation: Cardiovascular Genetics 10.2 (2017): e001605.)  

12.)Line 137: A citation is missing.

13.) DES (desmin) is also a LVNC gene and should be mentioned including a relevant citation:

Kulikova, Olga, et al. "The desmin (DES) mutation p. A337P is associated with left-ventricular non-compaction cardiomyopathy." Genes 12.1 (2021): 121.

Especially, because DES and LMNA are homologous IF proteins!

14.) Could you also add a short paragraph about restrictive cardiomyopathy (RCM) and LMNA mutations. It is known, that LMNA mutations cause also RCM.

Paller, Mark S., Cindy M. Martin, and Mary Ella Pierpont. "Restrictive cardiomyopathy: an unusual phenotype of a lamin A variant." ESC Heart Failure 5.4 (2018): 724-726.

15.) Please try to prevent to report unpublished data from your group. This is not fair, because the reviewer can not read the details to this point (Line 282).

16.) Could you describe novel techniques like genome-editing or base-pair editing, prime-editing in the context of laminopathies?

17.) Could you also summarize the existing animal models for LMNA?

18.) Could you please cite an article describing the LINC complex or could you prepare an overview figure.

In general, the reivew article is interesting but needs several extensions - especially relevant literature has not been cited.

However, I am optimistic that the authors can improve the quality of their review article in a major revision. Good luck!  

Author Response

In the review article 'Epigenetics in LMNA-related cardiomyopathy' submitted by Wang and Dobreva to Cells, the authors summarize the knowledge about LMNA mutations leading to cardiomyopathies. The topic of this review is interesting; however, the review article can be improved at different points.

We would like to express our sincere gratitude to the reviewer for his/hers constructive comments, as well as for the time invested in carefully reviewing our manuscript. The reviewer’s comments are in italics.

1.) Please prepare an overview figure about lamin A / C and Lamin B1 and B2. Please discuss important protein domains of these proteins.

Lines 33-40: We thank the reviewer for this comment. We have now included an overview figure and discussed the domain organization of nuclear lamins: “A-type and B-type lamins have a common structural organization: a short “head” domain at the N-terminus followed by a central helical rod domain and a C-terminal 'tail' domain. The central rod domain is composed of four coiled-coil regions that allows lamins to form parallel coiled-coil dimers and higher order meshworks [10-12]. The 'tail' consists of a globular region, which adopts an immunoglobulin (Ig)-like β-fold involved in protein-protein interactions. Pre-lamin A and B-type lamins also have a CaaX motif at the C-terminus which guides protein farnesylation and carboxyl methylation, important for targeting to the nuclear envelope [10-12] (Figure 1).

2.) Please explain inclusion and exclusion criteria for the LMNA-mutation table. This seems a pretty random selection and it is not complete. Human Gene Mutation Database analysis can be added.

We have highlighted the most studied LMNA mutations associated with DCM, LVNC, ARVC and RCM. In addition, we have included a reference to the UMD-LMNA mutation database (www.umd.be/LMNA), listing around 500 mutations and 300 protein variants.

3.).Line 93 ... develop DCM.  Please add a citation 

References to published work have been added (now line 124).

4.) Line 97 ... mutant mice. Please add a citation

Reference has been added (now line 128).

5.) Please list a relevant review article about the genetic background of DCM at the beginning of paragraph 2. For example you can cite the following book chapter: Gerull, Brenda, Sabine Klaassen, and Andreas Brodehl. "The genetic landscape of cardiomyopathies." Genetic Causes of Cardiac Disease (2019): 45-91.

Reference has been added in the beginning of paragraph 2 (line 70).

6.) Western blot and not western blot (Line 109).

Has been corrected according to the reviewers’ suggestion (now line 140).

7.) Could you please discuss the effect of proline on protein structure of lamin?

We have added a paragraph discussing the effect of proline on protein structure and phosphorylation of lamin A/C:

Lines 152-160: “Among laminopathy-associated missense mutations, gain of proline is the most common. Proline gain can significantly alter protein structure. For example, LMNA S143P missense mutation causes DCM and disturbs the coiled-coil domain, thus affecting lamin A/C assembly into the nuclear lamina. This results in nuclear fragility and reduced cellular stress tolerance [103]. Gain of proline might also affect protein phosphorylation through proline-directed kinases, such as the mitogen-activated protein (MAP) kinases, cyclin-dependent protein kinase 5 (CDK5), glycogen synthase 3, etc. Mutations resulting in gain of proline often result in striated muscle disease, suggesting a common underlying mechanism [104].”

8.) Please add at the beginning of paragraph 3 a relevant review article to give the reader an overview about the genetic background of ARVC --> Gerull, Brenda, and Andreas Brodehl. "Insights into genetics and pathophysiology of Arrhythmogenic cardiomyopathy." Current Heart Failure Reports 18 (2021): 378-390.

Reference has been added (line 163).

9.) Please write all human gene names in Capitals and Italics.

All human gene names have been corrected to capitals and italics.

10.) Line 125/126 Please add DSC2 encoding desmocollin-2. You should also insert a citation for DSC2. --> Brodehl, Andreas, et al. "A homozygous DSC2 deletion associated with arrhythmogenic cardiomyopathy is caused by uniparental isodisomy." Journal of Molecular and Cellular Cardiology 141 (2020): 17-29.

Reference has been added.

11.) You should also add the other non-desmosomal genes including the relevant citations in Line 127/128:

DES (Protonotarios, Alexandros, et al. "The novel desmin variant p. Leu115Ile is associated with a unique form of biventricular Arrhythmogenic Cardiomyopathy." Canadian Journal of Cardiology 37.6 (2021): 857-866.),

LEMD2 (Abdelfatah, Nelly, et al. "Characterization of a unique form of arrhythmic cardiomyopathy caused by recessive mutation in LEMD2." JACC: Basic to Translational Science 4.2 (2019): 204-221.),

PLN (Van Der Zwaag, P. A., et al. "Recurrent and founder mutations in the Netherlands—Phospholamban p. Arg14del mutation causes arrhythmogenic cardiomyopathy." Netherlands Heart Journal 21 (2013): 286-293.)

ILK (Brodehl, A., Rezazadeh, S., Williams, T., Munsie, N. M., Liedtke, D., Oh, T., ... & FORGE Canada Consortium. (2019). Mutations in ILK, encoding integrin-linked kinase, are associated with arrhythmogenic cardiomyopathy. Translational Research208, 15-29.)

and CDH2 (Mayosi, Bongani M., et al. "Identification of cadherin 2 (CDH2) mutations in arrhythmogenic right ventricular cardiomyopathy." Circulation: Cardiovascular Genetics 10.2 (2017): e001605.)  

Within the section “Arrhytomogenic cardiomyopathy”, we have included the references as follows:
Lines 165-173: “Mutations in desmosomal genes, such as Plakophilin 2 (PKP2), Desmoplakin (DSP), Desmoglein 2 (DSG2) , Desmocollin 2 (DSC2) and Junction plakoglobin (JUP), are the main cause of ARVC [107-112]. In addition, mutations of the calcium handling protein Ryanodine Receptor 2 (RYR2) [113], Phospholamban (PLN) [114], the adherens junction protein Cadherin 2 (CDH2) [115], Integrin-Linked Kinase (ILK)[116], the signaling molecule Transforming Growth Factor- β3 (TGFB3) [117] and the cytoskeletal structure protein, Titin (TTN) [118], as well as Desmin (DES) [119, 120], the Transmembrane protein 43 (TMEM43) and lamin A/C ( LMNA), have also been reported in ARVC [24, 121-123].”

12.) Line 137: A citation is missing.

Reference has been added (now line 183).

13.) DES (desmin) is also a LVNC gene and should be mentioned including a relevant citation:

Kulikova, Olga, et al. "The desmin (DES) mutation p. A337P is associated with left-ventricular non-compaction cardiomyopathy." Genes 12.1 (2021): 121.

Especially, because DES and LMNA are homologous IF proteins!

Reference has been added (lines 210-212).

14.) Could you also add a short paragraph about restrictive cardiomyopathy (RCM) and LMNA mutations. It is known, that LMNA mutations cause also RCM.

Paller, Mark S., Cindy M. Martin, and Mary Ella Pierpont. "Restrictive cardiomyopathy: an unusual phenotype of a lamin A variant." ESC Heart Failure 5.4 (2018): 724-726.

We have added a paragraph on restrictive cardiomyopathy as follows:

Lines 236-251: “5. Restrictive cardiomyopathy

Restrictive cardiomyopathy (RCM) is a rare cardiac disease characterized by increased myocardial stiffness resulting in impaired ventricular filling. Patients with RCM show enlarged atria and diastolic dysfunction, while systolic function and ventricular wall thicknesses are often normal until later stages of the disease [136-138]. Although most causes of RCM are acquired, several genes mutation have also been identified in patients with RCM [136-139]. The most common mutated genes found in RCM are sarcomere related genes such as TTN[140], TNNI3[141], MYH7[142], ACTC1[143], etc. Mutations of non-sarcomere genes such as DES[144], TMEM87B[145], FLNC[146] etc. have also been reported. Recently, Paller et al. reported a truncation mutation of LMNA (c.835 delG:p.Glu279ArgfsX201) in a RCM patient who had a significant biatrial enlargement, atrial fibrillation as well as skeletal muscle weakness. Both right and left ventricular size and function were normal and histological analysis revealed cardiac hypertrophy and focal interstitial fibrosis in the endomyocardial tissue [62]. How Lmna mutations cause RCM is not known, a plausible mechanism could be activation of profibrotic signaling as discussed below.”

15.) Please try to prevent to report unpublished data from your group. This is not fair, because the reviewer can not read the details to this point (Line 282).”

We have now deleted the sentence referring to unpublished work from our group.

16.) Could you describe novel techniques like genome-editing or base-pair editing, prime-editing in the context of laminopathies?

We thank the reviewer for this comment. Indeed, in the last years there has been a significant progress in genome editing of cardiomyopathies. Thus, we have added a paragraph: “Advances in therapeutic strategies for LMNA-related cardiomyopathy (lines 459-485)”.

17.) Could you also summarize the existing animal models for LMNA?

We have now included a new table (Table 2), summarizing the existing mouse models of Lmna-related cardiomyopathies.

18.) Could you please cite an article describing the LINC complex or could you prepare an overview figure?

We have now referred to three reviews focusing on the function of the LINC complex. In our review “The LINC Between Mechanical Forces and Chromatin” (ref. 21), there is an overview figure on this topic.

Lines 47-51: “In addition, lamins interact with the LINC complex, which couples the nucleoskeleton with the cytoskeleton [19, 20], and thereby can directly translate mechanical cues and changes in extracellular matrix mechanics, into alterations in chromatin structure and transcriptional activity [21].”

In general, the review article is interesting but needs several extensions - especially relevant literature has not been cited. However, I am optimistic that the authors can improve the quality of their review article in a major revision. Good luck!  

We thank the reviewer for the constructive comments and we sincerely hope that we have thoroughly answered all reviewers’ concerns.

Round 2

Reviewer 1 Report

The authors have fulfilled all the requests by this reviewer.

Reviewer 2 Report

The author list of reference 112 in the reference paragraph is wrong and has to be corrected ... The first author is Brodehl A and not AB, A. Please correct this.

All other points have been addressed correctly by the authors. 

Therefore I suggest a minor revision correcting the author(s) in reference 112.